# Integrated Multiresonator Quantum Memory

**DOI:** 10.3390/e25040623

**Published:** 2023-04-06

**Authors:** Nikolay Sergeevich Perminov, Sergey Andreevich Moiseev

**Affiliations:** 1Kazan Quantum Center, Kazan National Research Technical University, n.a. A.N.Tupolev-KAI, 10 K. Marx, 420111 Kazan, Russia; 2Zavoisky Physical-Technical Institute, Kazan Scientific Center of the Russian Academy of Sciences, 10/7 Sibirsky Tract, 420029 Kazan, Russia

**Keywords:** quantum information, multiresonator quantum memory, switchable coupler, integrated quantum device, spectrum optimization, spectral–topological matching condition

## Abstract

We develop an integrated efficient multiresonator quantum memory scheme based on a system of three interacting resonators coupled through a common resonator to an external waveguide via switchable coupler. It is shown that high-precision parameter matching based on step-by-step optimization makes it possible to efficiently store the signal field and enables on-demand retrieval of the signal at specified time moments. Possible experimental implementations and practical applications of the proposed quantum memory scheme are discussed.

## 1. Introduction

Quantum memory (QM) is of great importance for the successful development of quantum information technologies [1,2]. There are many promising approaches in the elaboration of efficient QM, intensively discussed in the reviews [3,4,5,6]. The use of resonators in the QM approaches being developed is attracting increasing attention due to the significant enhancement of the interaction of photons with emitters placed in the resonator and the possibilities of controlling these interactions [7,8,9,10,11,12,13,14]. Among the approaches, QM schemes using photon/spin echo on quantum information carriers with the inhomogeneous broadening of resonance transition open up promising possibilities in quantum storage of broadband multipulse fields [6]. Moreover, the stored signal pulses can have an arbitrary time mode, and their storage does not require strict time synchronization when using additional control fields. For example, this is the case for QM based on slow light [15,16]. The cavity-assisted photon echo QM, often called impedance-matched QM [11,12], has been demonstrated in a number of experiments on AFC and ROSE protocols [17,18,19,20,21], which show the possibility of implementing high efficiency, currently achieving up to 60%. Moreover, the impedance-matched approach significantly facilitates the requirements to the optical density of resonant transition and, consequently, expands the practical possibilities in the implementation of existing QM protocols, which has become especially important for the AFC protocol [19,20,22].

The cavity-assisted photon/spin echo QM has also begun to be used in the implementation of microwave QM [23,24,25]. However, it is not yet possible to achieve sufficiently high efficiency for broadband microwave pulses due to the need for low Q-factor resonators [26], where the interaction of electron spins with microwave photons becomes insufficiently strong [23]. In this regard, the use of resonators with high Q-factors seems to be important for significantly increasing the interaction of single photon fields with electron spins and facilitating methods of increasing the QM efficiency. However, an increase in the Q-factor of resonators reduces the operating spectral range of QM, which requires the search for new approaches that would allow for broadband QM based on the high-Q resonators. One of the approaches is using a high-Q resonator with zero spectral dispersion (so called white cavities) [22] and an alternative approach is based on the system of high-Q resonators whose natural frequencies can cover an arbitrarily wide spectral range [27].

The continuous improvement in the creation of various high-Q resonators and the development of integrated technologies has made it possible to create multiresonator systems that significantly facilitate the control of light fields [28], and has made such approaches promising for the realization of practically useful QM [27,29,30]. Moreover, in comparison with using an ensemble of electron spins in a typical microwave resonator, a system of high-Q resonators makes it possible to achieve a sufficiently strong interaction between such resonators for implementing the perfect dynamics of controlled broadband light fields. The first implementation of the echo-based memory protocol on a system of microwave resonators was demonstrated for intense microwave pulses [27] and showed the possibility of achieving perfect impedance matching of the multiresonator system with an external waveguide. Using the system of superconducting resonators has made it possible to significantly increase the Q-factor of resonators. In addition, the system of high-Q resonators with controlled frequencies allows implementing on-demand broadband QM [27], which was demonstrated experimentally on a chip of four superconducting resonators for microwave pulses attenuated to a single-photon level by Bao et al. [31]. The realized four-resonator QM has demonstrated the ability to store two pulses and rich functionality for the implementation of various controlled dynamics in transformation of quantum information and switching between different storage modes due to the ability to quickly change the frequencies of resonators. However experimentally achieved efficiency (6% in [31]) is still not high enough, which indicated the need to develop more effective ways to control the interaction of the multiresonator system with an external waveguide.

Recently the possibility of achieving high (60%) efficiency and accuracy in a quantum storage of microwave pulses attenuated to a single-photon level was shown when implementing multiresonator QM on a chip of superconducting resonators [32]. Moreover, the analysis of the results obtained in the work shows that the efficiency is limited only by the Q-factor of the resonators and the accuracy of setting their parameters, which allows us to conclude that it is possible to increase the efficiency of more than 90%, due to the broadband spectral-topological matching condition [33]. At the same time, the achieved efficiency was obtained for a fixed storage time of microwave pulses, determined by the inverse of the spectral interval between the nearest natural frequencies of a multiresonator system. However, a long-lived on-demand QM multiresonator should operate in two modes: storage/readout and free self-evolution without retrieval of radiation to an external waveguide. The second stage can be carried out in two ways. In the first case, this is achievable due to the freezing of macroscopic coherence in the resonator system by equalizing the frequencies of the internal resonators [27,31] during storage. In the second case, we can simply disconnect the multiresonator system from the external waveguide. The second method seems to be more preferable due to less influence on the Q-factor of the resonators, and on the quantum state of the radiation stored in the system of resonators that do not interact directly with an external waveguide. Below we are developing a second approach for on-demand multiresonator QM with a common resonator [27,32], which makes this approach particularly promising for the implementation of broadband QM.

Recently, applying the fast switch implemented in the work of Flurin et al. [34] to control communication with a superconducting microwave resonator, we have showed ([35]) the possibility of on-demand QM on a system of four resonators with a common resonator coupled with an external qubit (resonator) via a controlled switch. Perfect operation of this QM occurs when using the optimal parameters of the resonators ensures two conditions: (A)-the fulfillment of the impedance matching condition and (B)-preservatio of the periodic structure of resonant frequencies in the multiresonator system during transfer and storage stages of signal fields. In this work, as a first step we develop efficient on-demand multiresonator QM coupled via a controlled switch with external waveguide. The difference between this QM and the previously studied scheme in [35] is that the interaction with an external waveguide makes the considered resonator system an open (non-Hamiltonian) system where the interaction with the waveguide acquires a relaxation character which in another way affects the nature of spectral changes in the system of coupled resonators. Therefore, the direct application of the results obtained in the work [35] turns out to be impossible. Moreover, the solution that can ensure the preservation of both (A and B) conditions of the parameter matching becomes unobvious. Further, we show that in the studied system of three resonators interacting with one common resonator connected to waveguide via a switch, it is possible to find the optimal parameters of this system for implementing on-demand efficient QM. Used methods and solutions found are presented and the existing experimental possibilities of their implementation are discussed, including the optical frequency range based on using the latest achievements in the fabrication of high-Q resonators and fast switches.

## 2. Physical Model

Figure 1 shows a principal scheme of a multiresonator QM with three mini-resonators and one common resonator connected to an external waveguide via a switch. The mini-resonators have the following eigenfrequencies: $\omega_{n}=\omega_{0}+\Delta_{n}$, where $\Delta_{n}=\left\{-\Delta ,0,\Delta \right\}$ is the frequency offset of the side mini-resonators from the common resonator with central frequency $\omega_{0}$, and $f_{n}=f$ is the coupling constants of the mini-resonators with the common resonator. We assume that a switch can quickly change the coupling constant of the common resonator with the waveguide from κ=0 to constant $\kappa =\kappa_{0}$. Thus, we consider only two modes of operation when the common resonators have these two values of the coupling constant κ.

Taking into account the high Q-factor of the resonators, we neglect the field attenuation in the description of the dynamics of the studied QM at times *t*, assuming $t\ll 2Q⁄\omega_{n}$ (n={1,…,N}) and, using frequency units, we set the Hamiltonian in the form:(1)$$\begin{array}{c} H=\int d\omega \omega a_{\omega }^{†}a_{\omega }+\sum_{n}\omega_{n}b_{n}^{†}b_{n}+\omega_{0}a^{†}a+\left[\sum_{n}f_{n}a^{†}b_{n}+g_{cw}\int d\omega a_{\omega }^{†}a+h.c.\right], \end{array}$$
here $a^{†}$, *a* and $b_{n}^{†}$, $b_{n}$ are the creation and annihilation Bose operators of the mode of the common resonator and the mode of the *n*th mini-resonator ($\left[a,a^{†}\right]=1,\left[b_{n},b_{m}^{†}\right]=\delta_{n,m}$), respectively; $a_{\omega }^{†}$, $a_{\omega }$ are the operators of creation and annihilation of the ω-th mode of the waveguide ($\left[a_{\omega^{\prime }},a_{\omega }^{†}\right]=\delta \left(\omega^{\prime }-\omega \right)$ ), respectively.

We describe the quantum dynamics using the well-known input–output formalism of quantum optics [36] recently generalized to resonator–waveguide circuits [37] and well-recommended in the resonator microwave QMs [32,34].

We assume that all four resonators are prepared in the ground states $|G_{r}\rangle =\prod_{n=0}^{3}|g_{n}\rangle$ before signal pulses described by the initial state $|\psi_{in}\rangle$ are launched into the common resonator. Using Hamiltonian (Equation 1) and following the input–output approach [36] for the storage stage of an input signal field, we get the system of Langevin–Heisenberg equations for the resonator modes $a\left(t\right),b_{n}\left(t\right)$:(2)$$\begin{array}{cc} \left[\partial_{t}+\frac{\gamma_{n}}{2}+i\Delta_{n}\right]b_{n}+if_{n}a & =\sqrt{\gamma }F_{n},\\ \left[\partial_{t}+\frac{k+\gamma_{0}}{2}\right]a+i\sum_{n}f_{n}b_{n} & =\sqrt{k}a_{in}+\sqrt{\gamma_{0}}F_{0}, \end{array}$$
where we have taken into account a relaxation of the cavity modes with decay constants $\gamma_{n},\gamma_{0}$ and related Langevin Forces [38]: $F_{n}\left(t\right),F_{0}\left(t\right)$ ($\left[F_{m}\left(t\right),F_{m^{\prime }}^{†}\left(t^{\prime }\right)\right]=\delta_{m,m^{\prime }}\delta \left(t-t^{\prime }\right),k=2\pi g_{cw}^{2}$).

The input signal field $a_{in}\left(t\right)$ excites the common resonator, and the input- and outfields of the waveguide are coupled by the equation: $a_{in}\left(t\right)-a_{out}\left(t\right)=\sqrt{\kappa_{0}}a\left(t\right)$ [36]: ($a_{in,out}\left(t\right)=\frac{1}{\sqrt{2\pi }}\int d\omega e^{-i\omega t}{\tilde{a}}_{in,out}\left(\omega \right)$). The Equation (Equation 2) are found for the Fourier components:

$\left({\tilde{b}}_{n}\left(\omega^{\prime }\right),\tilde{a}\left(\omega^{\prime }\right)\right)=\frac{1}{\sqrt{2\pi }}\int dte^{i\omega^{\prime }t}\left(b_{n}\left(t\right),a\left(t\right)\right)$ and similar expressions for Langevin forces ${\tilde{F}}_{0}\left(\omega \right)$, ${\tilde{F}}_{n}\left(\omega \right)$ (where $\left[{\tilde{F}}_{n}^{†}\left(\omega^{\prime }\right),{\tilde{F}}_{m}\left(\omega \right)\right]=\delta_{m,m^{\prime }}\delta \left(\omega^{\prime }-\omega \right)$):(3)$$\tilde{a}\left(\omega \right)=2\frac{\sqrt{\kappa_{0}}{\tilde{a}}_{\mathrm{in}}\left(\omega \right)+\sqrt{\gamma_{0}}{\tilde{F}}_{\Sigma }\left(\omega \right)}{k+\gamma_{0}-2i\omega +2\chi \left(\omega \right)},$$
where χ(ω)=Re{χ(ω)}+i·Im{χ(ω)} is an effective permittivity of the memory with real and imaginary parts: (4)$$Re\left\{\chi \left(\omega \right)\right\}=\sum_{n=1}^{3}\frac{f_{n}^{2}\gamma_{n}/2}{{\left(\gamma_{n}/2\right)}^{2}+{\left(\Delta_{n}-\omega \right)}^{2}},Im\left\{\chi \left(\omega \right)\right\}=\sum_{n=1}^{3}\frac{f_{n}^{2}\left(\omega -\Delta_{n}\right)}{{\left(\gamma /2\right)}^{2}+{\left(\Delta_{n}-\omega \right)}^{2}}.$$

The solution (Equation 3) contains an effective Langevin operator ${\hat{F}}_{\Sigma }\left(\omega \right)$:(5)$${\tilde{F}}_{\Sigma }\left(\omega \right)={\tilde{F}}_{0}\left(\omega \right)-i\sum_{n=1}^{N}\frac{g_{n}\sqrt{\gamma_{n}/\gamma_{0}}}{\gamma_{n}/2+i\left(\Delta_{n}-\omega \right)}{\tilde{F}}_{n}\left(\omega \right),$$
that is essential for determining noise of the QM device at finite temperatures, where $\langle {\tilde{F}}_{n}^{†}\left(\omega \right),{\tilde{F}}_{n}\left(\omega \right)\rangle =\langle {\tilde{F}}_{0}^{†}\left(\omega \right),{\tilde{F}}_{0}\left(\omega \right)\rangle \approx n_{\mathrm{bath}}\left(\omega_{0}\right)$. Using (Equation 3) and (Equation 2) we get a solution for the mini-resonator modes:(6)$${\tilde{b}}_{n}\left(\omega \right)=\frac{-if_{n}\tilde{a}\left(\omega \right)+\sqrt{\gamma }{\tilde{F}}_{n}\left(\omega \right)}{\gamma /2+i(\Delta_{n}-\omega )},$$
and applying the relation between input and output fields we also find
(7)$$\begin{array}{cc} & {\tilde{a}}_{out}\left(\omega \right)=S\left(\omega \right)\cdot {\tilde{a}}_{in}\left(\omega \right)+{\tilde{b}}_{\mathrm{noise}}\left(\omega \right), \end{array}$$
(8)$$\begin{array}{cc} & {\tilde{b}}_{\mathrm{noise}}\left(\omega \right)=2\frac{\sqrt{k\gamma_{0}}{\tilde{F}}_{\Sigma }\left(\omega \right)}{k+\gamma_{0}+2\chi \left(\omega \right)-2i\omega }, \end{array}$$
where S(ω) is a spectral transfer function of the memory:(9)$$S\left(\omega \right)=\frac{k-\gamma_{0}-2\chi \left(\omega \right)+2i\omega }{k+\gamma_{0}+2\chi \left(\omega \right)-2i\omega },$$
${\tilde{b}}_{\mathrm{noise}}\left(\omega \right)$ is a noise component in the output signal caused by the interaction with the bath modes of all the resonators. The noise spectrum Equation (8) (see also Equation (Equation 5)) contains a wide line (with a spectral width ∼κ) strongly suppressed by the factor $\frac{\gamma_{0}}{k}\ll 1$ due to the coupling with the external waveguide and a periodic comb of narrow lines (with a spectral width ∼γ) caused by a quantum noise in the mini-resonators. The contribution of the noise of the mini-resonators to the signal field decreases by ∼γ/Δ≪1, which indicates a weakened influence of the quantum noise of a multiresonator system on the state of the radiation stored in it. The transfer function S(ω) is characterised by eigenfrequencies of the resonator system which are highly sensitive to the interaction between the resonators modes. Recently [33], it was found that the interaction greatly changes the arrangement of the eigenfrequencies $\omega_{n}^{\prime }$. The behavior of the eigenfrequencies experiences a topological transition near the impedance matching condition, while the frequency arrangement should take the form of a periodic frequency structure to ensure high memory efficiency, similar to a photon echo on atomic frequency combs (AFC) [12,17,39,40]. Moreover high efficiency is also possible under more general conditions [35] when the eigenfrequencies $\omega_{n}^{\prime }$ become a multiple of a certain frequency, for example $\omega_{n}^{\prime }=\left[-4,-1,1,4\right]\cdot \Delta^{\prime }$, arranged symmetrically relative to zero frequency offset, where $\Delta^{\prime }$ becomes a function of several parameters $\Delta^{\prime }=\Delta^{\prime }\left(\Delta ,f,k\right)$.

Highly efficient quantum storage means the realization of an almost perfect delay of the signal field for a given time interval *T* ($T=\frac{2\pi }{\Delta^{\prime }}$) so that $S\left(\omega \right)≅e^{i\omega T}$ in the operating frequency range with spectral efficiency $E\left(\omega \right)={\left|S\left(\omega \right)\right|}^{2}≅1$. This condition also means minimization of the reflected field $\langle a_{out}\left(t\right)\rangle ≅0$ and emptying the common resonator a(t<T)≅0. At the same time, the search for optimal parameters of a multiresonant system should be carried out taking into account the behavior of the stored signal when the resonators are disconnected from the external waveguide. Assuming instantaneous disconnection of the resonators at a time $t=t_{0}$, we obtain the following system of equations for the resonator modes for $t>t_{0}$:(10)$$\begin{array}{cc} \partial_{t}b_{n}\left(t\right)+if_{n}a\left(t\right)= & \left[\delta \left(t-t_{0}\right)-\left(\frac{\gamma }{2}+i\Delta_{n}\right)\right]b_{n}\left(t\right)+\sqrt{\gamma }F_{n}\left(t\right)\eta \left(t-t_{0}\right),\\ \partial_{t}a\left(t\right)+i\sum_{n}f_{n}b_{n}\left(t\right)= & \left[\delta \left(t-t_{0}\right)-\frac{\gamma_{0}}{2}\right]a\left(t\right)+\sqrt{\gamma_{0}}F_{0}\left(t\right)\eta \left(t-t_{0}\right). \end{array}$$
where $\eta (t-t_{0})$ is a Heaviside function ($\eta (t-t_{0})=1$ for $t>t_{0}$; and =0 for $t<t_{0}$).

In the general case, the Equation (Equation 10) lead to the appearance of new eigenfrequencies in the resonator system and these frequencies will not necessarily form a periodic structure of resonant lines necessary to ensure high efficiency of recovery of the stored signal. To increase the efficiency and operational functionality of the multiresonator circuit, we apply methods of fast precision control and periodic positioning of the natural frequencies of coupled resonators together with impedance matching, using fast algorithms of algebraic and numerical optimal control [33]. Of great interest is the situation when the spectral width of the signal pulse becomes comparable with the spectral width of the comb of the natural frequencies of the resonators. Here we also show that the efficiency can be further improved by specializing the spectral profile of the field that carries quantum information between elements of a quantum device.

## 3. Efficiency Optimization Procedure

In order to achieve efficient QM in the proposed topology of resonators connection, it is required to optimize the controlled QM parameters in two operating modes. Switching between them is carried out by controlling the connection of the common resonator with the external waveguide. The Langevin forces contained in the Equations (Equation 2), (Equation 9) and (Equation 10) do not affect the efficiency of QM and therefore will not be considered further below. Assuming the presence of a periodic structure of the initial frequencies of three miniresonators (see Figure 1), we start searching for the optimal coupling constant *f* of these resonators with a common resonator that is disconnected from an external waveguide k=0 where the distribution of coupling constants is chosen in the form $\left[f_{1},f_{2},f_{3}\right]=f\cdot \left[0.8,1,0.8\right]$ for the possibility of obtaining an equidistant spectrum. This situation corresponds to the long-term storage stage (p=2) of signal radiation in the multiresonator system, which requires a multiplicity of natural frequencies of the system, at which it becomes possible to periodically restore the stored quantum state in time. For equal decay constants ($\gamma_{0}=\gamma_{n}=\gamma$), from the Equation (Equation 9) we get the eigenfrequencies $\omega_{n}^{\left(p\right)}={\tilde{\omega }}_{n}^{\left(p\right)}-i\gamma$ of the multiresonator system (p=2: k=0) where ${\tilde{\omega }}_{n}^{\left(p\right)}$ are the solutions of the following algebraic equation:(11)$$\begin{array}{c} S_{11}\left(\omega \right)=-\frac{k\omega \left(\Delta^{2}-\omega^{2}\right)-2iP\left(\omega \right)}{k\omega \left(\Delta^{2}-\omega^{2}\right)+2iP\left(\omega \right)},\\ P\left(\tilde{\omega }\right)={\tilde{\omega }}^{4}-\left(f_{1}^{2}+f_{2}^{2}+f_{3}^{2}+\Delta^{2}\right){\tilde{\omega }}^{2}+\Delta^{2}f_{2}^{2},\\ P(\tilde{\omega })=0. \end{array}$$

The behavior of the eigenfrequencies ${\tilde{\omega }}_{n}^{\left(2\right)}$, depending on the constant coupling *f*, is shown in Figure 2. It is worth noting that with the growth of the coupling constant *k* of the common resonator with an external waveguide, when certain critical values are reached ($k_{1}\approx 7$, $k_{2}\approx 5.5$), the two eigenfrequencies merge into one [33] and remain unchanged for each of these two cases considered (see Figure 2). This behavior characterizes a kind of topological transition found earlier for such systems in our work [33]. From the obtained numerical dependencies, we find two sets of multiple eigenfrequencies: ${\tilde{\omega }}_{n}^{\left(2\right)}\left(f_{1}\right)=\left[-4,-1,1,4\right]\cdot \Delta^{\prime }\left(f_{1}\right)$ (where new spectral interval $\Delta^{\prime }\left(f_{1}\right)=0.529$) at a coupling constant $f_{1}=1.119$) that reproduces results of work [35] and ${\tilde{\omega }}_{n}^{\left(2\right)}\left(f_{2}\right)=\left[-3,-1,1,3\right]\cdot \Delta^{\prime }\left(f_{2}\right)$ (where $\Delta^{\prime }\left(f_{2}\right)=0.589$ at a coupling constant $f_{2}=1.038$). The choice of such eigenfrequencies makes it possible to read the stored state out at multiple intervals of time $T_{n}\left(f_{1}\right)=2\pi n/\Delta^{\prime }\left(f_{1}\right)$ or $T_{n}\left(f_{2}\right)=2\pi n/\Delta^{\prime }\left(f_{2}\right)$.

Assuming a given coupling constant $f_{1}$ or $f_{2}$, at the second step we can look for the optimal value of the constant coupling $k=\kappa (f_{n})$ of the common resonator with an external waveguide. Similar to work [27], at this stage of optimization, we impose a smoothness condition on the phase delay τ(ω)=−iArg[S(ω)]/ω at the point ω=0, which can be represented as $\partial_{\omega =0}^{2}\left[\tau \right]=0$, where for near zero losses during the interaction with signal pulse temporal duration δt ($\gamma_{n}\delta t\ll 1,\gamma_{0}\delta t\ll 1$) in the resonator’s transfer function $S_{11}\left(\omega \right)$ defined by the Equation (Equation 11):(12)$$\begin{array}{c} \tau \left(\omega \right)=\frac{2}{\omega }\arctan\left[\frac{k\omega (\Delta^{2}-\omega^{2})}{2(\omega^{4}-\left(f_{1}^{2}+f_{2}^{2}+f_{3}^{2}+\Delta^{2}\right)\omega^{2}+\Delta^{2}f_{2}^{2})}\right]. \end{array}$$

The transfer function $S_{11}\left(\omega \right)$ describes the spectral properties of the reflected signal. Strong suppression of the reflected signal in a certain spectral interval corresponds to the implementation of the impedance-matched QM [11,12]. Where from which we obtain the following smoothness condition for the coupling constants k=κ and $f_{n}$ in algebraic-type form:(13)$$\begin{array}{c} \partial_{\omega =0}^{2}\left[\tau \right]=0,\\ \kappa =\sqrt{\frac{\Delta^{2}f_{2}^{2}+\left(f_{1}^{2}+f_{2}^{2}+f_{3}^{2}\right)\left(f_{1}^{2}+f_{3}^{2}\right)}{\Delta^{2}/12}}. \end{array}$$

This condition is necessary for the complete loading of the signal from the external waveguide into the QM. For $f_{n}=1.119\cdot \left[1,1,1\right]$ and multiplicity spectra type [−4,−1,1,4] we will get $\kappa_{1}=7.256$, for $f_{n}=1.038\cdot \left[0.8,1,0.8\right]$ and equidistant spectra type [−3,−1,1,3] we will get $\kappa_{2}=5.546$. Figure 3 shows a phase delay τ(ω) with a smoothed region near the central frequency and which has a plateau corresponding to the operating spectral range of QM. To implement the memory, we use the operating relative frequency range ω=[−0.5,0.5], where the corresponding high quantum fidelity of signal reconstruction is greater than 0.9.

For the case of coupling (p=1: k=κ) (Equation 13) in equation $S^{-1}\left(\omega_{n}^{\left(1\right)}\right)=0$, we find the dependence of four eigenfrequencies $Re(\omega_{n}^{\left(1\right)})$ from the coupling constant *f* and, similarly, we find the dependence $Re(\omega_{n}^{\left(1\right)})$ from the constant *g*. By assuming the equal decay constants ($\gamma_{0}=\gamma_{n}=\gamma$), we get the eigenfrequencies $\omega_{n}^{\left(p\right)}={\tilde{\omega }}_{n}^{\left(p\right)}-i\gamma$ of the multiresonator system for the two cases (p=1: k=κ and p=2: k=0) where ${\tilde{\omega }}_{n}^{\left(p\right)}$ are the solutions of the following algebraic equation:(14)$$2f^{2}\left(\Delta^{2}-3{\tilde{\omega }}^{2}\right)-\tilde{\omega }\left(\Delta^{2}-{\tilde{\omega }}^{2}\right)\left(2\tilde{\omega }+ik\right)=0,$$
where we also assume that the coupling constant k(t) changes fast enough between its two values k=κ and k=0 that the evolution of radiation during the switching time $\delta t_{s}$ when dk/dt≠0 can be neglected.

The frequencies $\omega_{n}^{\left(1\right)}$ determine the poles of the transfer function S(ω), describing the stationary absorption spectrum of a loaded multiresonator system, while $\omega_{n}^{\left(2\right)}$ are the eigenfrequencies of a disconnected system where $Im[\omega_{n}^{\left(2\right)}]=0$ if γ=0. In Figure 2 the eigenfrequencies are presented for both cases (p=1,2), where we see that when the coupling constant k(t) is changed between the two states (k=κ and k=0), the energy spectrum and, accordingly, the internal dynamics of the multiresonator system changes significantly. Thus, a natural problem is matching the energy spectra of the system in different operating modes with on and off coupling, in which it will be possible to synchronize the dynamics of the two stages of evolution, allowing both the effective loading of signal pulse and its perfect on-demand retrieval after long-term storage. In the proposed scheme, this problem is solvable due to the controlled flexible spectral characteristics of the QM with a common central resonator.

## 4. Recording Dynamics

Due to the fact that there is a time interval near $t=T_{0}/2$, when almost all the energy of the initial signal pulse is transferred in mini-resonators, and in the presence of a rapidly switching coupling k(t), QM can be transferred at this moment of time into operation for long-term storage. In this case, the switching process can be noiseless due to the fact that there is almost no energy in the common resonator and waveguide. After several cycles in the mode of long-term storage of a signal pulse in a system of high-Q resonators, which has reversible temporal dynamics, we can similarly turn on the k(t) coupling and on-demand readout the signal pulse into an external waveguide.

To implement the optical version of the proposed QM, high-Q resonators can be used, which have a high coupling strength between each other [41], due to which it becomes possible to achieve high efficiency of information storage during one cycle η=exp{−2γ/f}>0.9, where f>200 MHz and γ<10 MHz are experimentally achievable coupling parameters between resonators and decay constant. For such values of the coupling constants, we constructed a characteristic intensity curve for the echo in the waveguide for a Gaussian-like profile signal field with half-width σ≈1 in the presence of loading, one cycle of storage and unloading of the signal (corresponds to the distribution of resonator excitation in the middle of the storage cycle of the form $x_{n}\left(t=T_{storage}/2\right)=\left[1,-2,1\right]$). Figure 4 shows the dependence of the relative signal intensity in the outer waveguide in time representation, where there is one cycle of the storage phase with disconnected communication, corresponding to a plateau in the central time region. To achieve a higher efficiency of the proposed memory scheme [33], additional optimization methods can be used, which is the subject of study in the following works.

## 5. Discussion

We have demonstrated the ability to control two different regimes of efficient quantum storage in a multiresonator QM connected with waveguide via switchable coupling for the possibility of integrating with other superconducting circuits. The optimal parameters of this QM are found, from which it becomes possible to efficiently load signal pulses and store them for an arbitrary time, which is limited only by the intrinsic Q-factor of the resonators used. It is noteworthy that the most optimal coupling constants of a multiresonator QM with an external waveguide are established near the region of the topological phase transition, the location of which depends on the magnitude of the coupling constants. Interestingly, the presence of this type of spectral–topological phase transition is associated with the impedance matching condition. Thereby it becomes possbile to implement a periodic or multiple structure of eigenfrequencies in the multiresonator QM.

The proposed QM mutiresonator can be implemented for microwave fields in a system of high-Q coplanar superconducting resonators, where, due to the broadband spectral matching, a record efficiency of more than 60% and a fidelity close to 100% have recently been demonstrated for the microwave pulse attenuated to a single-photon level [32]. The main technological advantage of the proposed QM circuit is its compactness due to a small number of mini resonators, and of possibly achieving large interaction constants between the resonators with an external waveguide. It is also worth noting the possibility of quickly adjusting the parameters of a multiresonator system to the operating mode thanks to the use of the optimal algebraic relations found. This tuning capability is a unique feature of multiresonator quantum memory circuits and provides multifrequency broadband matching of the spectral characteristics of quantum memory and the stored signal in a large spectral range [27], in contrast to other QM circuits on resonators.

To implement a multiresonator QM in the optical frequency range, it is necessary to solve a technological problem of combining several controlled subsystems on a single chip. One of the promising approaches to solve this problem is the use of silicon planar photonics [42], where it is possible to achieve high efficiency of energy transfer between the elements to build the high-performance quantum devices. To provide an efficient quantum storage of quantum signal in a many-particle system, where high accuracy of parameter control is needed, it seems possible to use the recently appeared new high-performance components, such as nonophotonic waveguide [43] and microresonators with ultra-high Q-factor [41,44,45,46]. Thus, the optical implementation of the proposed multiresonator QM is technologically possible, which, however, is not a task that can be solved in one step.

The microresonators can be efficiently and accurately controlled [47,48,49] and combined with resonant long-lived single atoms [50] for a significant increasing storage time. The proposed multiresonator QM characterized by high-Q factor can be an efficient interface providing strong interaction of photons with the long-lived carrier of quantum information. This approach makes it possible to realize both a high storage efficiency of quantum signals and good control over the dynamic characteristics of the very dynamic process of storing quantum information. In this regard, we show that, by optimizing the parameters of a many-particle system, a small number of resonators can be used in this configuration to implement the highly efficient storage of quantum information, which greatly simplifies the technological implementation. It is worth noting that involvement of long-lived atoms in a system of high-Q resonators will require the search for additional methods for controlling atomic coherence and the use of quantum transitions of three- and multi-level quantum systems with different character of inhomogeneous broadening [18,51,52,53,54] will be important in the implementation of the multiresonator echo based QM protocols.

## Figures and Tables

**Figure 1 entropy-25-00623-f001:**
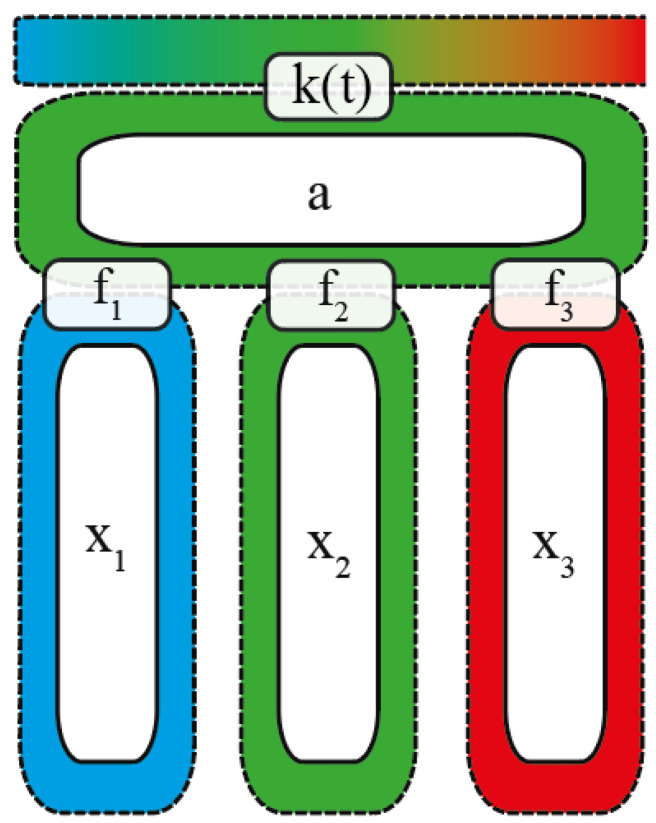
Principal scheme of a multiresonator QM with switcher, which is integrated with an external waveguide.

**Figure 2 entropy-25-00623-f002:**
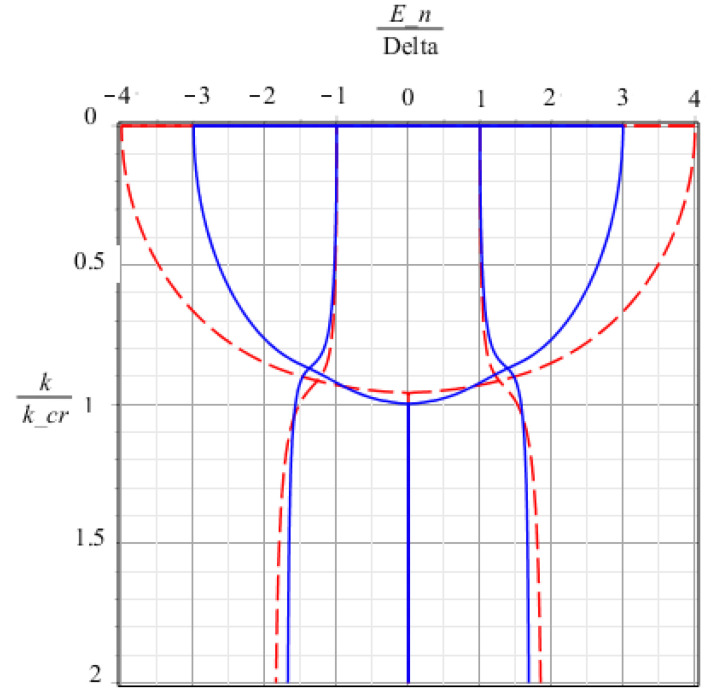
QM eigenfrequencies $E_{n}=Re\left(\omega_{n}\right)$ for $f_{n}=1.119\cdot \left[1,1,1\right]$—dashed line, $f_{n}=1.038\cdot \left[0.8,1,0.8\right]$—solid line.

**Figure 3 entropy-25-00623-f003:**
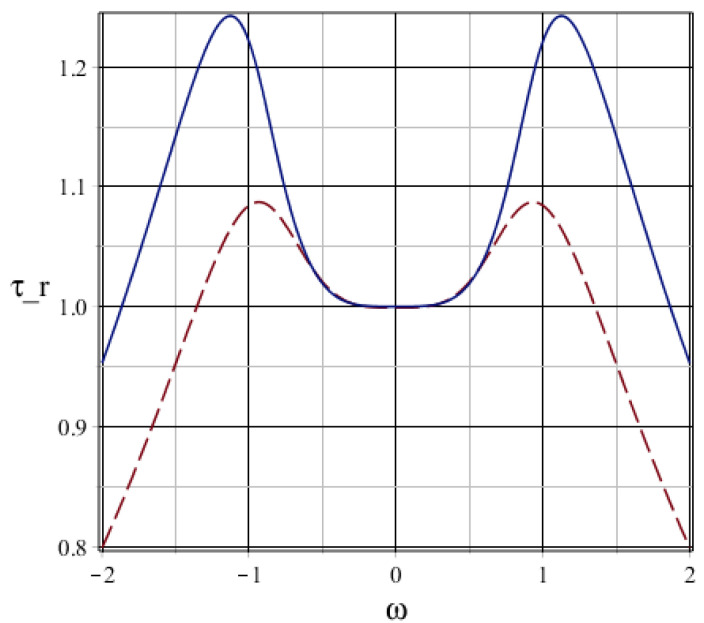
Relative phase delay $\tau_{r}=\tau \left(\omega \right)/\tau \left(0\right)$ for multiplicity spectra type [−4,−1,1,4]—dashed line, and equidistant spectra type [−3,−1,1,3]—solid line.

**Figure 4 entropy-25-00623-f004:**
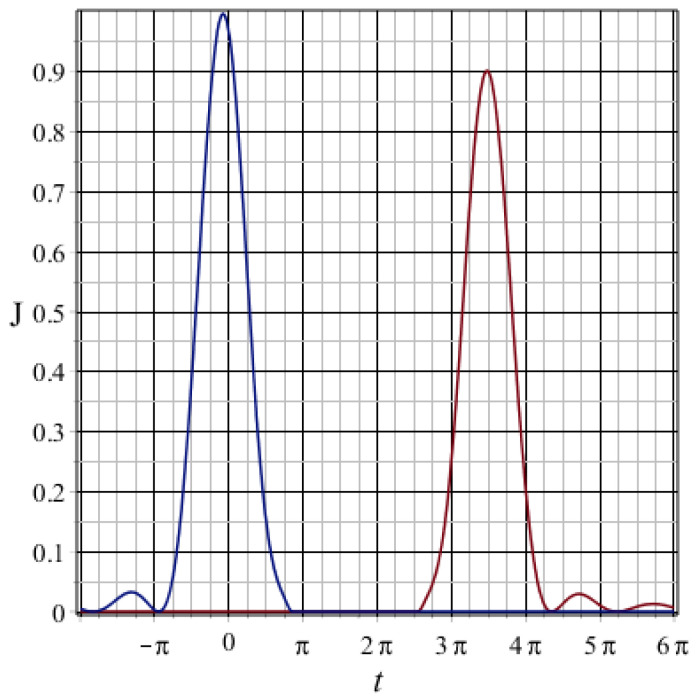
Relative intensity J(t) of the echo signal for a Gaussian-type pulse with half-width σ≈1 for equidistant spectra type [−3,−1,1,3].

## Data Availability

Data sharing are not applicable.

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
