# Peer review of "Integrated Multiresonator Quantum Memory"

_entropy, 2023, doi:10.3390/e25040623_

Round 1

Reviewer 1 Report

The work by N. S. Perminov and S. A. Moiseev, entitled “Integrated multiresonator quantum memory,” describes a scheme for the implementation of a quantum memory using three interacting resonators coupled to a common one. The latter is then coupled to an external waveguide, which is used to drive the micro-resonator system.

I believe the work is well-written and technically sound. However, in its present form, the manuscript seems more like a mathematical exercise than a serious experimental proposal. It is not clear whether the values chosen for the resonator-frequencies, couplings, readout times, etc., give the best results or they represent the actual values that can be accessed experimentally. I would like the authors to discuss this point in detail.

The proposed quantum-memory scheme is assumed to use signals in the microwave regime. Here I would like the authors to discuss how easy it would be to actually use the stored information to, say, communicate at long distances. It is well known that transduction is a complicated process that could represent an overall decrease in the quantum memory efficiency. It would be good to discuss the overall efficiency of microresonator-based memories when compared to atomic or rare-earth-doped-crystal-based quantum memories.

Related to my previous point, the authors discuss the possibility of implementing their scheme using optical frequencies; however, as far as I understand, Ref. [41] deals with the design and implementation of single resonators not arrays of them. References [42] and [43] describe the coupling of the resonators to a waveguide, but again arrays are not demonstrated. I encourage the authors to revise the last paragraph of Section 4, and the third paragraph of Section 5; I believe the potential readers need to know that the implementation of the proposed scheme is not a straightforward task.

Once the above mentioned issues have been addressed, I believe the manuscript can be considered for publication in Entropy.

Author Response

Dear Reviewer!

We attached the revised manuscript with our answers to yours comments.

Sincerely,

N.Perminov and S.Moiseev

Reviewer 2 Report

This paper proposes a scheme to build an optical Quantum Memory (QM), appropriate to operate with microwaves. In principle, this QM would work also for fields attenuated to a single photon. The device consits of 4 cavities (resonators). One main cavity is coupled to the waveguide, by one side, and to 3 other cavities on the other side (Fig. 1). The main cavity can be switched on to and off from the waveguide, controlling storage and transmission of the information (microwave pulse) in the memory.

The bulk of the paper is dedicated to calculate the optimal parameters (coupling constants) for the operation of the QM. They start with the Hamiltonian in Eq. 1 and then solve the Langevin- Heisenberg equations for the resonator modes (Eq.2). The optimal parameters are obtained by means of numerical calculations.

I wouldn’t dare to say I have checked all the calculations, which are also beyond my expertise! But the results make sense!

(Please notice an “n” missing in the equation at line 84.)

In the conclusion, the authors mention that the proposed QM can achive efficiency of more than 60% and almost perfect fidelity.

The device seems to be of low cost (just cavities!), compared to other QMs. I hope this QM gets off the ground.

The paper is written in a highly technical style, wich limits the range or readers. I think that besides improving the english, the authors could be a little bit more pedagogical in the description of the calculations, aiming non specialists.

I recommend the publication of this paper.

Author Response

(The authors gave the same response as above.)

Round 2

Reviewer 1 Report

The authors have properly addressed all my concerns. I believe the manuscript can now be published in Entropy.

Reviewer 2 Report

I am pleased with the revised version.